# CLIP-Map: Structured Matrix Mapping for Parameter-Efficient CLIP Compression

## Abstract

Contrastive Language-Image Pre-training (CLIP) has achieved widely applications in various computer vision tasks, e.g., text-to-image generation, Image-Text retrieval and Image captioning. However, CLIP suffers from high memory and computation cost, which prohibits its usage to the resource-limited application scenarios. Existing CLIP compression methods typically reduce the size of pretrained CLIP weights by selecting their subset as weight inheritance for further retraining via mask optimization or important weight measurement. However, these select-based weight inheritance often compromises the feature presentation ability, especially on the extreme compression. In this paper, we propose a novel *mapping-based* CLIP compression framework, **CLIP-Map**. It leverages learnable matrices to map and combine pretrained weights by *Full-Mapping with Kronecker Factorization*, aiming to preserve as much information from the original weights as possible. To mitigate the optimization challenges introduced by the learnable mapping, we propose *Diagonal Inheritance Initialization* to reduce the distribution shifting problem for efficient and effective mapping learning. Extensive experimental results demonstrate that the proposed CLIP-Map outperforms select-based frameworks across various compression ratios, with particularly significant gains observed under high compression settings.

## 1 Introduction

Large-scale language-image pre-training models, e.g., CLIP (Radford et al., 2021), achieve outstanding zero-shot transfer capability, which has been widely applied to various computer vision tasks, such as, text to image generation (Rombach et al., 2022) and scene understanding (Gu et al., 2021; Rao et al., 2022; Liu et al., 2023). However, such models are accompanied by large parameters and heavy computation costs, which restrict their real-world applications and deployments.

Pruning (Frankle & Carbin, 2018) and knowledge distillation (Hinton et al., 2015; Sanh et al., 2019) are two commonly used techniques for compressing multimodal models. Pruning can be broadly divided into two categories. The first is token pruning (Rao et al., 2021; Bolya et al., 2022; Shi et al., 2023b; Cao et al., 2024), which aims to reduce the computational cost (FLOPs) by selecting and removing less informative tokens during inference. The second is model pruning (Zhou et al., 2021; Sun et al., 2023; Frantar & Alistarh, 2023), which focuses on reducing the number of model parameters and consequently reduces FLOPs. In this paper, our discussion focuses on the model pruning methods. Several previous studies (Shi et al., 2023a; Wu et al., 2023; Lin et al., 2024) have extensively explored how to effectively compress CLIP-like models using a combination of pruning and knowledge distillation. Most of these approaches adopt a pruning-retraining pipeline that first selects and prunes unimportant parameters and then applies retraining to recover performance. The key difference between these methods lies in the different important weight measurement methods to select, with various strategies (Shi et al., 2023a; Lin et al., 2024; Wu et al., 2023) proposed to assess the importance of each parameter.

Regardless of the specific pruning strategy employed, pruning is a *select-based* method and inevitably leads to information loss from the pretrained model. Even with the retraining phase, it remains challenging to recover the information loss caused by dropping unimportant parameters or tokens.

Weight initialization plays a critical role in the training process of neural networks. A well-designed initialization scheme (Glorot & Bengio, 2010; He et al., 2015) can lead to more stable training process

and improved final performance. Recent studies (Chen et al., 2015; 2021; Wang et al., 2023a;c; Xia et al., 2024) have begun to explore initialization schemes with model growth paradigms, where a larger model is initialized by inheriting and expanding a pretrained smaller one, aiming to fully transfer the knowledge of smaller pretrained model to a larger one, bring a better initialization and accelerate the training process of large models. Among these, LiGO (Wang et al., 2023a) represents a prominent line of *mapping-based* work that reformulates model growth as an optimization problem by introducing learnable mapping parameters. These mapping parameters are optimized to search the most effective expansion strategy, enabling efficient transfer of knowledge from the small to the large model. LeTs (Xia et al., 2024) extends the work of LiGO by introducing a compact learnable module called **LearnGene** (Wang et al., 2022a; 2023b), along with a set of learnable transformations. Given a desired target model size, the corresponding transformation is applied to the LearnGene module to produce initialization weights for a model of arbitrary size. This design enables scalable and adaptive initialization across a wide range of model capacities.

Inspired by these works, we aim to combine these works by replacing traditional select-based compression strategies with a learnable mapping-based paradigm, construct a **mapping-retraining** pipeline to acquire a compact compressed model with fewer information loss. However, directly transferring learnable mapping techniques from model expansion to model compression presents several challenges. Firstly, almost all model growth techniques use a partial mapping and weight inheritance schemes, where a subset of the pretrained smaller model parameters is copied to the larger model and learns a mapping for the remaining part of the larger model. However, in model compression scenarios, the target parameter matrix is typically smaller than the original one, making such a partial mapping and weight inheritance approach inapplicable. Secondly, mapping a larger model into a smaller one requires a substantial number of parameters, which introduces significant overhead in both storage and computation, while also increasing the complexity of the optimization space. Thirdly, mapping-based methods have been mainly studied in the context of unimodal architectures, leaving the mapping of multimodal models such as CLIP partly unexplored.

To solve these limitations, we propose *CLIP-Map*, a mapping-retraining multimodal model compression pipeline. Our CLIP-Map first acquires a better initialization of compressed model using learnable mappings, and then retrain the initialized model using knowledge distillation. Specially, we use learnable parameter-efficient matrices $F^{in}$ and $F^{out}$ obtained by Kronecker Factorization to map large model parameters blocks into smaller counterparts using matrix multiplication. And using $L_{depth}$ to linear combination different layers and get a model with fewer layers. Due to the effectiveness of weight inheritance proved by prior works (Chen et al., 2021; Sanh et al., 2019; Wu et al., 2023), we initialize $F^{in}$ and $F^{out}$ as diagonal matrices. This initialization method ensures that part of original pretrained parameters is copied after multiplying with $F^{in}$ and $F^{out}$, enabling easier optimization. After initializing in this way, we then optimize them to find the best mapping structure. We introduce knowledge distillation in the retraining stage, using the cross-entropy loss between the logits of the student model and those of the teacher model as soft labels, enabling the student to mimic the behavior of the teacher and effectively inherit its knowledge.

We apply our mapping-based method to models of varying scales and find superior performance compared to select-based method such as TinyCLIP of similar size across multiple benchmarks. Notably, our method maintains competitive performance even under extremely high compression ratios. Moreover, our approach requires fewer training epochs, highlighting its efficiency in both performance and training cost.

In summary, our contributions include:

1. We propose a mapping-based compression method that, unlike conventional select-based pruning approaches, avoids hard parameter removal and better preserves the full information contained in the pretrained model and acquires a better-initialized compact model.

2. We replace the the select-based pruning method in the pruning-retraining pipeline with our mapping-based compression method, constructing a simplified mapping-retraining pipeline with less engineering complexity and better performance.

3. Our method maintains strong performance even under high compression ratios with fewer training epochs, demonstrating its effectiveness and efficiency in extreme compression scenarios.

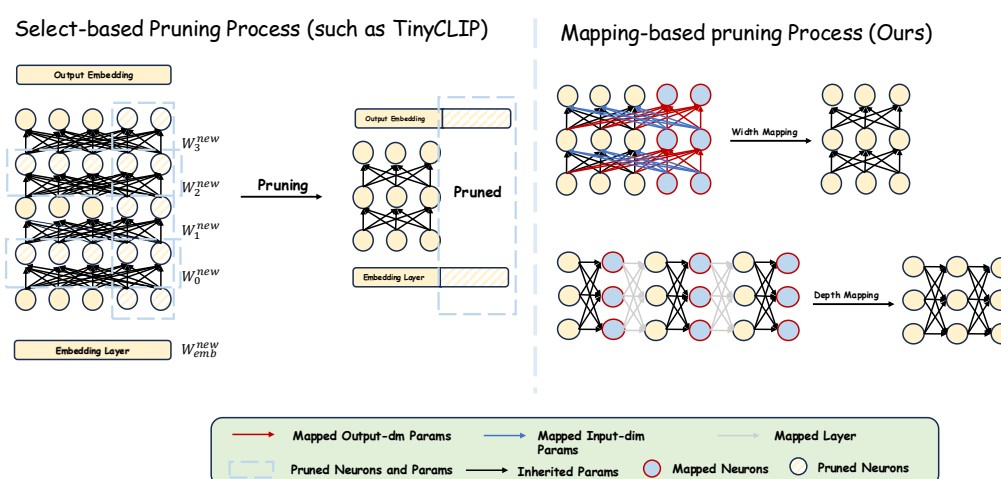

Figure 1: Select-based Compression Method and Mapping-based Growth Method.

## 2 RELATED WORK

### 2.1 MODEL COMPRESSION AND ACCELERATION FOR CLIP

There are many techniques exploring model compression and acceleration, including quantization Krishnamoorthi (2018); Wu et al. (2020), pruning Frankle & Carbin (2018); Sun et al. (2023) and knowledge distillation Hinton et al. (2015); Jiao et al. (2019); Wu et al. (2022); Touvron et al. (2021), etc. With the surge in multimodal foundation models such as CLIP Radford et al. (2021), several studies Gan et al. (2022); Shi et al. (2023a); Touvron et al. (2021) have begun to investigate how these established compression and acceleration techniques on unimodal models can be effectively adapted to the multimodal setting. Unlike unimodal architectures, multimodal models introduce unique challenges such as cross-modal alignment and modality-specific structures, making the direct application of traditional compression methods nontrivial. As a result, recent efforts focus on designing compression frameworks that jointly preserve both modality-specific capabilities and cross-modal interactions.

In the context of CLIP compression and acceleration, the most commonly adopted techniques are pruning and knowledge distillation. A common approach to pruning involves introducing a learnable or heuristic mask to measure the importance of individual weights or channels within the network. The mask can guide us to select a more compact sub-net of the original model, inheriting most orignal model's information. Specifically, Gan et al. (2022) shows that the lottery ticket hypothesis (Frankle & Carbin, 2018), originally proposed for unimodal models, still holds in multimodal settings. Regarding inter-modality dependencies, UPop (Shi et al., 2023a) performs joint pruning across both visual and textual modalities. Fig. 1 left provides a simple illustration of the select-based pruning process. Building on pruning, other works (Wang et al., 2022b; Wu et al., 2023; Lin et al., 2024) incorporate knowledge distillation to mitigate information loss introduced by pruning and leverage a powerful teacher model to enhance student performance. Further, CLIP-KD (Yang et al., 2024) extends the work of TinyCLIP(Wu et al., 2023) by investigating an empirical study of CLIP model distillation and identifies optimal design choices for distilling CLIP models. There is another line of pruning work (Rao et al., 2021; Shi et al., 2023b; Cao et al., 2024) focusing on token pruning by eliminating redundant and less informative tokens during the forward pass to achieve inference acceleration. Despite reducing computational overhead during inference through fewer computing tokens, token pruning does not decrease the model's parameter numbers and may, in some instances, incur additional parameter overhead due to the inclusion of pruning modules.

### 2.2 GROWING PRETRAINED MODELS FOR BETTER INITIALIZATION AND EFFICIENT TRAINING

To mitigate the cost of training deep neural networks, recent approaches (Chen et al., 2015; Gong et al., 2019; Chen et al., 2021; Wang et al., 2023a) explore model growth techniques, which initializes

a larger model using a pretrained smaller model of the same architecture, aiming to transfer the knowledge of pretrained smaller model to the larger model, enabling more efficient training and faster convergence. Net2Net (Chen et al., 2015) introduces function-preserving initialization, which guarantees that a larger model initialized from a smaller one produces identical outputs for the same inputs, thereby preserving and transferring the learned knowledge smoothly. StackBERT (Gong et al., 2019) found that the outputs of different Transformer layers exhibit similarity to some extent. Based on this observation, it proposes to grow the model by simply duplicating a subset of layers from a pretrained BERT (Devlin et al., 2019) and stacking them to form a deeper model. LiGO (Wang et al., 2023a) and LeTs (Xia et al., 2024) formulate model growth as a learnable optimization problem, where a smaller model is transformed into a larger one through parameterized mapping functions as in Fig. 1 right. These mappings are jointly optimized to preserve knowledge while adapting to the increased model capacity.

Our approach is inspired, but distinguished from previous work in the following three major aspects:

- **Mapping-based Compression.** Unlike mapping-based model growth methods, our work focus on model compression using this mapping method.

- **Multimodal Adaptation.** In contrast to previous mapping-based growth methods that are primarily applied to unimodal architectures such as BERT, our approach is specifically designed for multimodal vision-language models like CLIP.

- **Unified and Simplified Pipeline.** Unlike methods that decouple width and depth compression into multiple stages or rely on handcrafted pruning strategies, our framework introduces a unified, end-to-end optimization pipeline. This design simultaneously learns the width and depth compression mappings in a fully differentiable manner, reducing engineering complexity while achieving superior compression-performance trade-offs.

## 3 METHOD

### 3.1 PRELIMINARIES AND NOTATIONS

**Notation.** Consider a neural network with $L_1$ layers, each with a hidden dimension of $D_1$. Let the weight matrix at layer $l$ be denoted as $\boldsymbol{W}_l \in \mathbb{R}^{D_1 \times D_1}$. We use an operator $\boldsymbol{R}_l$ to acquire a smaller counterpart using matrix multiplication

$$Vec(\boldsymbol{W}_l^{'}) = \boldsymbol{R}_l Vec(\boldsymbol{W}_l) \in \mathbb{R}^{D_2 \times D_2}, \boldsymbol{R}_l \in \mathbb{R}^{D_2^2 \times D_1^2}. \tag{1}$$

Here, $Vec$ is an operator flattening a parameter matrix into a column vector. We combine each layer's $\boldsymbol{R}_l$ to get the width-compression operator $\boldsymbol{R}_{width}$.

For depth compression, we use a depth-compression operator $\boldsymbol{L}_{depth}$ to acquire a shallower network with $L_2$ layers, where each new layer can be expressed as a linear combination of old layers as in Eq. 2.

$$\boldsymbol{W}_{l'}^{\text{new}} = \sum_{l=1}^{L_1} \boldsymbol{L}_{depth}[l', l] \cdot \boldsymbol{W}_l, \quad l' = 1, \dots, L_2; \boldsymbol{L}_{depth} \in \mathbb{R}^{L_2 \times L_1}. \tag{2}$$

### 3.2 THE PROPOSED CLIP-MAP

#### 3.2.1 OVERVIEW

To compress the CLIP model, we need to introduce the operators $\boldsymbol{R}_{width}$ and $\boldsymbol{L}_{depth}$ operators for the text and visual encoder respectively. Since both modalities share a similar transformer (Vaswani et al., 2017) structure, we focus our discussion on the text encoder for notational simplicity.

We display our mapping-retraining pipeline in Fig. 2. In mapping stage, we freeze original large CLIP model and train the mapping parameters for both image encoder and text encoder. After mapping stage, we acquire a compressed CLIP model inheriting part of the original model. In the second retraining stage, we use this model as initialization of the student model and reuse the original model as a teacher model to distill the student model.

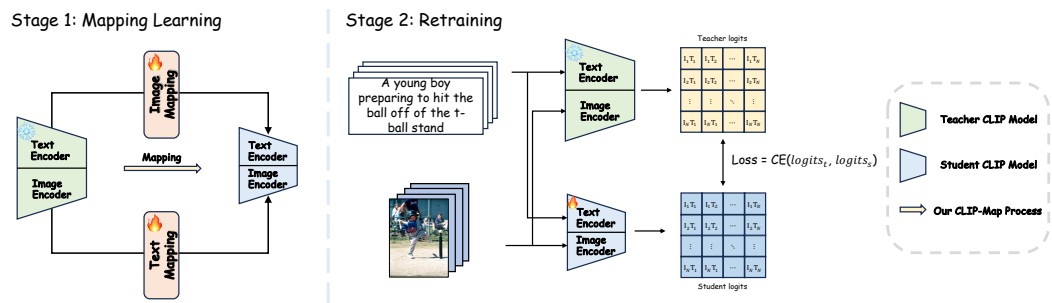

Figure 2: In the mapping-learning stage, we freeze original model's parameters and train mapping parameters only. In the retraining stage, we use knowledge distillation to distill the student model initialized by mapping stage.

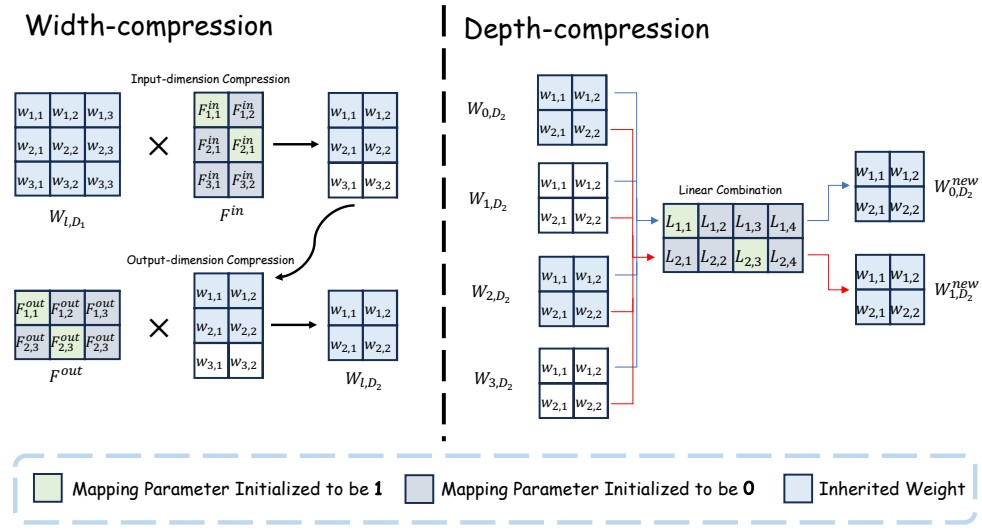

Figure 3: We firstly perform width-compression in both input-dimension and output-dimension on each layers parameter blocks. Then, we perform depth-compression to linear combining the compressed parameter blocks to a new layer parameter block.

In the following sections, we first introduce the core components used in the mapping stage: **Full-Mapping with Kronecker Factorization** and **Diagonal Inheritance Initialization**, which are detailed in Sec 3.2.2 and 3.2.3, respectively. Then, in Sec 3.2.4, we describe the design and training strategy of the retraining stage.

### 3.2.2 FULL-MAPPING WITH KRONECKER FACTORIZATION

As shown in Fig. 3, we apply the Full-Mapping with Kronecker Factorization strategy to compress a pretrained CLIP model. For the parameter block in $l$-th layer $\boldsymbol{W}_{l,D_1} \in \mathbb{R}^{D_1 \times D_1}$, our target size is $\boldsymbol{W}_{l,D_2} \in \mathbb{R}^{D_2 \times D_2}$, where $D_2 < D_1$. To acquire $\boldsymbol{W}_{l,D_2}$, the simplest approach is using $\boldsymbol{R}_l$ and mapping $\boldsymbol{W}_{l,D_1}$ using Eq. 1. which leads to the number of parameters $\mathcal{O}(D_1^2 D_2^2)$. By leveraging the property of Kronecker products, $\boldsymbol{R}_l Vec(\boldsymbol{W}_{l,D_1})$ can be reformulated as:

$$Vec(\boldsymbol{W}_{l,D_2}) = (\boldsymbol{F}_l^{in} \otimes \boldsymbol{F}_l^{out}) Vec(\boldsymbol{W}_{l,D_1}), \tag{3}$$

$$(\boldsymbol{F}_l^{in} \otimes \boldsymbol{F}_l^{out}) Vec(\boldsymbol{W}_{l,D_1}) = Vec(\boldsymbol{F}_l^{out} \boldsymbol{W}_{l,D_1} \boldsymbol{F}_l^{in^\top}), \tag{4}$$

where $\otimes$ denotes the Kronecker product and $\boldsymbol{F}_l^{in}, \boldsymbol{F}_l^{out} \in \mathbb{R}^{D_2 \times D_1}$. Therefore, we only use two trainable parameter matrices, $\boldsymbol{F}_l^{in}$ and $\boldsymbol{F}_l^{out}$, reducing the parameter scale from $\mathcal{O}(D_1^2 D_2^2)$ to $\mathcal{O}(D_1 D_2)$.

Due to the property of the Kronecker product of Eq. 4, it is unnecessary to explicitly construct the full mapping matrix. Instead, we treat the Kronecker factors $\boldsymbol{F}_l^{in}$ and $\boldsymbol{F}_l^{out}$ as transformations along the input and output dimensions, respectively, and apply them sequentially through standard matrix multiplication.

### 3.2.3 DIAGONAL INHERITANCE INITIALIZATION

During training of mapping, we observe that it is difficult to optimize the mapping matrices $\boldsymbol{F}_l^{in}$ and $\boldsymbol{F}_l^{out}$. Since $\boldsymbol{F}_l^{in}$ and $\boldsymbol{F}_l^{out}$ are derived from a Kronecker decomposition of the original mapping matrix $\boldsymbol{R}_{width}$, directly initializing them using common strategies(e.g. Xavier (Glorot & Bengio, 2010) and Kaiming (He et al., 2015) initialization) often results in *distribution shifting* problem, leading to numerical instability and poor convergence behavior.

**Formulation.** Consider $\boldsymbol{A} \in \mathbb{R}^{D_2 \times D_1}$ and $\boldsymbol{B} \in \mathbb{R}^{D_2 \times D_1}$ being two matrices independently initialized. Their Kronecker product is denoted as $\boldsymbol{R} = \boldsymbol{A} \otimes \boldsymbol{B} \in \mathbb{R}^{D_2^2 \times D_1^2}$. For any element of $\boldsymbol{R}$, we have:

$$\boldsymbol{R}_{(i-1)D_2+k,\ (j-1)D_1+l} = \boldsymbol{A}_{ij} \cdot \boldsymbol{B}_{kl}, \tag{5}$$

where $1 \leq i, k \leq D_2$ and $1 \leq j, l \leq D_1$.

Assuming each element of $\boldsymbol{A}$ and $\boldsymbol{B}$ is sampled independently from a zero-mean distribution with variance $\sigma_A^2$ and $\sigma_B^2$ respectively, we obtain:

$$\mathbb{E}[\boldsymbol{R}_{ij}] = \mathbb{E}[\boldsymbol{A}_{ij} \cdot \boldsymbol{B}_{kl}] = \mathbb{E}[\boldsymbol{A}_{ij}]\,\mathbb{E}[\boldsymbol{B}_{kl}] = 0, \tag{6}$$

$$\mathrm{Var}(\boldsymbol{R}_{ij}) = \mathrm{Var}(\boldsymbol{A}_{ij} \cdot \boldsymbol{B}_{kl}) = \mathbb{E}[\boldsymbol{A}_{ij}^2]\,\mathbb{E}[\boldsymbol{B}_{kl}^2] = \sigma_A^2 \cdot \sigma_B^2. \tag{7}$$

This implies that although the Kronecker-structured mapping $\boldsymbol{R}$ remains zero-mean under independent initialization, its variance becomes multiplicative in nature. If $\boldsymbol{A}$ and $\boldsymbol{B}$ are both initialized using schemes with moderate variance, the resulting transformation $\boldsymbol{R}$ may have a significantly shifting variance:

$$\mathrm{Var}(\boldsymbol{R}) = \sigma_A^2 \cdot \sigma_B^2. \tag{8}$$

Such uncontrolled variance scaling can lead to unstable optimization dynamics, including gradient vanishing or explosion during the early stage of training. Therefore, careful design or reparameterization of the initialization process is required to ensure stable convergence.

Weight inheritance has been proved to be an effective initialization strategy that facilitates knowledge transfer and accelerates convergence during training (Chen et al., 2021; Sanh et al., 2019; Wu et al., 2023). Thus, we propose **Diagonal Inheritance Initialization**. As illustrated in Eq. 4, the Kronecker factorization can be interpreted as a compression scheme that operates along both the input (in-dim) and output (out-dim) dimensions. Therefore, if we initialize the diagonal elements of both $\boldsymbol{F}_l^{in}$ and $\boldsymbol{F}_l^{out}$ to 1, it effectively preserves part of the original parameter structure as shown in Fig. 3, thus implementing weight inheritance with much less distribution shifting. Specially, we initialize the diagonal entries of both $\boldsymbol{F}_l^{in}$ and $\boldsymbol{F}_l^{out}$ and set the off-diagonal elements to zero or small random values

$$(\boldsymbol{F}_l^{in})_{ij}, (\boldsymbol{F}_l^{out})_{ij} = \begin{cases} 1, & \text{if } i = j, \\ 0, & \text{otherwise} \end{cases} \tag{9}$$

This structured initialization ensures that the initial Kronecker product approximates an identity-like transformation, i.e.,

$$\boldsymbol{R}_{width} \approx \boldsymbol{I}, \tag{10}$$

thus preserving the original parameter semantics in early training. The initialization strategy for $\boldsymbol{F}_{in}$ and $\boldsymbol{F}_{out}$ across different components of the network is illustrated in detail in A.3.

### 3.2.4 RETRAINING STAGE

In the retraining stage, we introduce knowledge distillation. A CLIP model first extract image embedding and text embedding of a image-text pair using image encoder and text encoder, and then compute image-text similarity(logits). We can use teacher logits as soft labels, forcing student model to mimic the distribution of teacher logits using cross-entropy loss (Wu et al., 2023; Yang et al., 2024).

$$\mathcal{L}_{distill} = CE(logits_{I2T}^s, logits_{I2T}^t) + CE(logits_{T2I}^s, logits_{T2I}^t). \tag{11}$$

Table 1: Zero-shot image-text retrieval results on MSCOCO and Flickr30K datasets (TR@K: image-to-text retrieval, IR@K: text-to-image retrieval). Our method achieves strong performance with fewer parameters. "2x25ep/3x25ep" in † denotes the method employs the two-stage/three-stage progressive compression strategy, each stage containing 25 training epochs. Detailed architectural configurations for all models can be found in A.3 Tab. 6.

| Method | Params (M) | Training Dataset | MSCOCO | | | | | | Flickr30K | | | | | |
|---|---|---|---|---|---|---|---|---|---|---|---|---|---|---|
| | | | TR@1 | TR@5 | TR@10 | IR@1 | IR@5 | IR@10 | TR@1 | TR@5 | TR@10 | IR@1 | IR@5 | IR@10 |
| SLIP (Mu et al., 2022) | 86+38 | YFCC-15M | 31.1 | - | - | 20.3 | - | - | 57.6 | - | - | 40.1 | - | - |
| CLIP (Wu et al., 2023) | 86+38 | YFCC-15M | 26.5 | - | - | 17.1 | - | - | 51.6 | - | - | 32.2 | - | - |
| CLIP (Radford et al., 2021) | 86+38 | WIT-400M | 49.6 | 74.0 | 82.1 | 30.1 | 54.3 | 65.1 | 79.2 | 95.0 | 98.0 | 59.2 | 83.2 | 89.6 |
| OpenCLIP (Cherti et al., 2023; Ilharco et al., 2021) | 86+38 | LAION-2B | 60.0 | 82.6 | 89.1 | 41.3 | 66.5 | 76.2 | 86.2 | 98.0 | 99.5 | 69.8 | 90.4 | 94.6 |
| MetaCLIP (Xu et al., 2023) | 86+38 | - | 58.6 | 81.0 | 88.5 | 41.0 | 66.7 | 76.6 | - | - | - | - | - | - |
| CLIP (Radford et al., 2021) | 38+38 | WIT-400M | 49.3 | - | - | 30.7 | - | - | 78.1 | - | - | 59.2 | - | - |
| CLIP (Radford et al., 2021) | 86+38 | WIT-400M | 50.8 | - | - | 28.3 | - | - | 81.2 | - | - | 58.2 | - | - |
| *Training data: YFCC15M* | | | | | | | | | | | | | | |
| *Compression Ratio: 1.0%* | | | | | | | | | | | | | | |
| TinyCLIP (Wu et al., 2023) | 0.8+0.3 | YFCC15M | 10.5 | 26.1 | 36.2 | 5.9 | 16.9 | 25.1 | 21.3 | 43.3 | 55.6 | 11.8 | 30.2 | 40.4 |
| † TinyCLIP (Wu et al., 2023) (3×25ep) | 0.8+0.3 | YFCC15M | 12.5 | 29.3 | 39.0 | 6.9 | 19.2 | 28.1 | 24.5 | 48.7 | 60.6 | 14.6 | 34.7 | 46.5 |
| **CLIP-Map_tiny (Ours)** | 0.8+0.3 | YFCC15M | **15.8** | **34.7** | **45.3** | **8.2** | **22.3** | **31.7** | **30.3** | **56.6** | **66.3** | **17.9** | **40.9** | **52.6** |
| *Compression Ratio: 10.0%* | | | | | | | | | | | | | | |
| TinyCLIP (Wu et al., 2023) | 8+3 | YFCC15M | 33.8 | 59.6 | 70.4 | 20.2 | 42.9 | 54.6 | 60.3 | 84.0 | 89.8 | 40.2 | 69.0 | 78.7 |
| † TinyCLIP (Wu et al., 2023) (2×25ep) | 8+3 | YFCC15M | 36.2 | 62.6 | 72.4 | 21.5 | 45.4 | 57.6 | 62.3 | 86.1 | 92.2 | 42.3 | 71.5 | 80.9 |
| **CLIP-Map_small (Ours)** | 8+3 | YFCC15M | **38.4** | **64.6** | **74.4** | **24.3** | **48.4** | **59.9** | **66.0** | **88.5** | **92.6** | **46.9** | **73.9** | **82.6** |
| **CLIP-Map_small (Ours, Meta-CLIP)** | 8+3 | YFCC15M | 34.3 | 60.1 | 70.8 | 22.6 | 46.5 | 58.5 | - | - | - | - | - | - |
| *Compression Ratio: 50.0%* | | | | | | | | | | | | | | |
| TinyCLIP (Wu et al., 2023) | 39+19 | YFCC15M | 54.9 | 79.4 | 87.2 | 38.9 | 64.2 | 74.1 | 84.6 | 96.7 | 99.0 | 66.7 | 88.7 | 93.7 |
| **CLIP-Map_base (Ours)** | 39+19 | YFCC15M | **55.1** | 78.8 | 86.5 | 37.9 | 63.8 | 74.1 | 81.9 | 96.2 | 98.8 | **67.6** | **89.0** | **94.0** |
| **CLIP-Map_base (Ours, Meta-CLIP)** | 39+19 | YFCC15M | 53.0 | 76.9 | 85.1 | 37.1 | 63.4 | 73.7 | - | - | - | - | - | - |
| **CLIP-Map_base (Ours, ResNet-50, wo Retraining)** | 19+19 | YFCC15M | 25.5 | 49.4 | 61.4 | 14.3 | 34.0 | 45.8 | - | - | - | - | - | - |

Table 2: Zero-shot classification top-1 accuracy on 21 downstream datasets. CLIP-Map exhibits strong performance across most tasks.

| Method | Image Encoder | Food101 | CIFAR10 | CIFAR100 | SUN397 | Stanford Cars | FGVC Aircraft | VOC2007 | DTD | Oxford Pets | Caltech101 | Flowers102 | MNIST | STL10 | EuroSAT | RESISC45 | GTSRB | KITTI | Country211 | PCam | Rendered SST2 | ImageNet-1K |
|---|---|---|---|---|---|---|---|---|---|---|---|---|---|---|---|---|---|---|---|---|---|---|
| *Zero-shot performance* | | | | | | | | | | | | | | | | | | | | | | |
| CLIP (Radford et al., 2021) | ViT-B/16 | 85.0 | 88.2 | 62.0 | 63.2 | 57.5 | 20.0 | 78.0 | 40.4 | 85.2 | 81.6 | 65.3 | 69.5 | 97.6 | 43.1 | 53.5 | 41.2 | 31.5 | 20.2 | 60.2 | 60.5 | 64.4 |
| OpenCLIP (Cherti et al., 2023; Ilharco et al., 2021) | ViT-B/16 | 86.6 | 94.9 | 76.8 | 70.8 | 88.5 | 27.0 | 78.8 | 56.3 | 90.5 | 83.8 | 71.3 | 65.8 | 97.9 | 53.4 | 62.8 | 48.2 | 17.0 | 20.3 | 56.4 | 59.9 | 70.2 |
| †TinyCLIP (Wu et al., 2023) | ViT-0.8M/16 | 22.2 | 31.2 | 11.5 | 30.1 | 2.2 | 3.0 | 44.6 | 11.3 | 17.4 | 40.5 | 34.4 | 11.4 | 63.8 | 16.5 | 12.4 | 28.0 | 4.9 | | 50.0 | 49.3 | 16.6 |
| **CLIP-Map_tiny(Ours)** | ViT-0.8M/16 | **24.5** | **41.9** | **13.6** | **32.7** | 2.1 | **3.2** | 38.6 | **13.7** | **18.8** | **41.2** | **37.7** | 10.7 | **72.4** | 8.4 | **14.9** | 3.9 | **32.5** | 5.5 | **50.1** | **50.0** | **19.0** |
| †TinyCLIP (Wu et al., 2023) | ViT-8M/16 | 59.6 | 72.8 | 42.1 | 56.4 | 7.7 | 6.9 | 62.0 | 28.8 | 46.2 | 71.7 | 58.0 | 9.8 | 92.5 | 23.3 | 20.7 | 10.8 | 15.3 | 11.7 | 52.8 | 50.0 | 41.1 |
| **CLIP-Map_small(Ours)** | ViT-8M/16 | **62.8** | **77.7** | **45.4** | **57.6** | **10.7** | **9.6** | **68.5** | **30.0** | **50.9** | **71.9** | 57.7 | 9.8 | 92.3 | **29.5** | **27.3** | **11.0** | 13.2 | **12.5** | **57.4** | **50.1** | **42.7** |
| TinyCLIP (Wu et al., 2023) | ViT-39M/16 | 82.7 | 91.3 | 67.7 | 69.2 | 51.7 | 15.1 | 76.9 | 47.3 | 80.8 | 81.9 | 70.0 | 38.0 | 97.3 | 52.4 | 54.9 | 31.6 | 11.1 | 18.3 | 60.5 | 49.9 | 63.5 |
| **CLIP-Map_base(Ours)** | ViT-39M/16 | 82.7 | **91.4** | **68.3** | 69.2 | 50.8 | **22.2** | **77.0** | **48.5** | **83.1** | 81.4 | **70.2** | 13.0 | 97.3 | **52.6** | **55.1** | 27.0 | 10.3 | **18.6** | 51.7 | 48.7 | **63.7** |

For hard label, we adopt the InfoNCE loss (He et al., 2020; Radford et al., 2021) used in standard CLIP training.

$$\mathcal{L}_{task} = CE(logits^s_{I2T}, labels) + CE(logits^s_{T2I}, labels). \tag{12}$$

The overall training loss is a weighted sum of hard task loss and distillation soft loss, with the weighting controlled by a coefficient $\lambda$.

$$\mathcal{L}_{total} = (1-\lambda)\mathcal{L}_{task} + \lambda\mathcal{L}_{soft}. \tag{13}$$

## 4 EXPERIMENT

### 4.1 EXPERIMENTAL SETTINGS

**Architecture.** We adopt a Transformer-base (Vaswani et al., 2017) CLIP architecture and design three variants of our model—**CLIP-Map_base**, **CLIP-Map_small** and **CLIP-Map_tiny**—each tailored to different application scenarios and resource constraints. CLIP-Map_base, CLIP-Map_small and CLIP-Map_tiny are mapped and distilled from OpenCLIP-ViT-B/16 (Cherti et al., 2023; Ilharco et al., 2021). We also evaluate our approach on Meta-CLIP (Xu et al., 2023) and CLIP using ResNet (He et al., 2016) as vision encoder to validate the generalization capability of our approach to any CLIP-like architecture. It should be noted that when ResNet serves as the vision encoder, we limit the process to the Mapping stage 5-epochs training, and don't perform the subsequent Retraining stage.

**Training setup.** We use YFCC-15M (Li et al., 2021), a large-scale public image-text pair datasets for training. The overall training procedure is organized into two stages: mapping stage and retraining stage. During mapping stage, we optimize learnable mapping matrices, and in retraining stage, we use each mapped model' s original model as a teacher model to distill the mapped model. All models are trained on 32 Nvidia H800. Detailed training settings are presented in A.5.

Table 3: Zero-shot IN-val performance comparison of various compressed CLIP models under different datasets.

| Methods | Dataset | # of Seen Samples | Params(M) *img+txt* | Zero-shot IN-val |
|---|---|---|---|---|
| †TinyCLIP-0.8M/16 | YFCC-15M | 1.125B | 0.8 + 0.3 | 16.6 |
| **CLIP-Map**tiny | YFCC-15M | 0.45B | 0.8 + 0.3 | 19.0 |
| †TinyCLIP-8M/16 (Wu et al., 2023) | YFCC-15M | 0.75B | 8 + 3 | 41.1 |
| ViT-T/16 (Yang et al., 2024) | CC3M (Sharma et al., 2018) + CC12M (Changpinyo et al., 2021) | 0.48B | 5.6 + 21.3 | 42.6 |
| **CLIP-Map**small | YFCC-15M | 0.45B | 8+3 | 42.7 |
| MobileCLIP-S0 (Vasu et al., 2024) | DataCompDR-12M | 0.74B | 11.4 + 42.4 | 59.1 |
| MoPE-CLIPbase (Lin et al., 2024) | YFCC-15M | 0.30B | 86+42 | 60.7 |
| TinyCLIP-39M/16 Wu et al. (2023) | YFCC-15M | 0.75B | 39+19 | 63.5 |
| **CLIP-Map**base | YFCC-15M | 0.30B | 39+19 | **63.7** |

Table 4: Effect of different initialization steps on downstream performance. Longer initialization improves accuracy across ImageNet-1K classification (IN-1K) and MSCOCO retrieval.

| Init. Steps | IN-1K Top-1 (%) Acc. | MSCOCO TR@K | | | MSCOCO IR@K | | |
|---|---|---|---|---|---|---|---|
| | | TR@1 | TR@5 | TR@10 | IR@1 | IR@5 | IR@10 |
| Manual Drop (0 epoch) | 41.1 | 33.8 | 59.6 | 70.4 | 20.2 | 42.9 | 54.6 |
| 0.28(1000steps) + 25 epochs | 39.7 | 35.2 | 61.1 | 71.3 | 21.7 | 44.6 | 56.4 |
| 1 + 24 epochs | 39.6 | 35.7 | 61.5 | 71.6 | 21.0 | 44.3 | 56.4 |
| 3 + 22 epochs | 41.9 | 37.6 | 63.1 | **73.9** | 23.0 | 46.9 | 58.7 |
| 5 + 20 epochs | **42.1** | **38.3** | **63.6** | **73.9** | **23.1** | **47.1** | **59.2** |
| 7 + 18 epochs | 40.8 | 36.2 | 61.9 | 72.3 | 22.0 | 45.5 | 57.6 |

**Benchmarks and Metrics.** We evaluate our method on two tasks: zero-shot classification and zero-shot retrieval. For the retrieval task, we adopt the MSCOCO (Lin et al., 2014) and Flickr30K (Plummer et al., 2015) test sets, using top-k image-to-text and text-to-image recall rates(TR@$K$ and IR@$K$) as evaluation metrics. For the classification task, we mainly conduct evaluations on the ImageNet-1K (Deng et al., 2009) test set, and report top-1 accuracy(Acc@$1$) to measure model performance. Beyond ImageNet-1K, we further evaluate the classification capabilities of our model on a diverse set of downstream zero-shot classification tasks.

## 4.2 MAIN RESULTS

**Zero-shot Image-text Retrieval.** Tab. 1 presents the zero-shot retrieval performance of our CLIP-Maptiny, CLIP-Mapsmall and CLIP-Mapbase models on the MSCOCO (5k test set) and Flickr30k(1k test set). To enable a fair comparison, we replicate three TinyCLIP models of equivalent size using the official TinyCLIP approach, under both progressive and non-progressive compression settings. For the progressive compression strategy, we start from the original OpenCLIP-ViT-B/16 and progressively compress it to 50.0%, 10.0%, and 1.0% of its original size. Models obtained via this progressive compression are marked with a † symbol in the result table. Under the compression ratio of 10.0% and 1.0%, our CLIP-Mapsmall and CLIP-Maptiny consistently outperform TinyCLIP of the same size across all recall metrics. And our CLIP-Mapbase achieves competitive performance but with fewer training epochs.

**Zero-shot Classification.** Tab. 2 shows the zero-shot classification performance of our method on multiple classification datasets. Compared to TinyCLIP models of similar sizes, our approach demonstrates competitive performance at the *base* scale, achieving results comparable to the baseline. Notably, under the *small* and *tiny* configurations, our model consistently outperforms the baseline across the majority of classification benchmarks, with substantial performance gains observed on several datasets.

**Comparison with Other VLM Compression Methods.** As shown in Tab. 3, we compare our method with other CLIP compression methods. The effectiveness of different approaches is evaluated via zero-shot classification accuracy on ImageNet-1K, while the overall efficiency is assessed based on the total volume of seen samples during the entire training pipeline and the model size. Compared to MoPE-CLIP (Lin et al., 2024) and CLIP-KD (Yang et al., 2024), our methods can get a better performance even with smaller model size and fewer seen samples, highlighting our method in both effectiveness and efficiency. MobileCLIP (Vasu et al., 2024) leverages an augmented dataset, DataCompDR (Gadre et al., 2023; Vasu et al., 2024), constructed via advanced data augmentation techniques. Compared to commonly used datasets such as YFCC-15M and CC3M, DataCompDR

offers higher data quality, thereby enabling MobileCLIP to achieve superior accuracy even with a significantly smaller image encoder. The practical training speed-up brought by our method over TinyCLIP is visualized and presented in A.6. Additional comparative results with other compression methods can be found in A.4.

### 4.3 ABLATION STUDIES

To demonstrate the effectiveness of our method, we conduct ablation studies for different initialization methods and mapping/retraining duration. We also investigate the effect of training loss in A.8.

**Different Initialization Methods.** To validate the effectiveness of *Diagonal Inheritance Initialization*, we conduct experiments on the OpenCLIP-ViT-B/16 model under a 10.0% compression ratio, comparing the training process and performance of different initialization strategies for the mapping parameter matrices. We adopt Random init, Kaiming init (He et al., 2015) and Xavier init (Glorot & Bengio, 2010) for comparison. The loss curves and final performance of different initialization methods are illustrated in A.7 Fig. 6 and summarized in Tab. 5. The *Diagonal Inheritance Initialization* lead to a easier optimization and faster convergence loss curve, bringing much better performance on test sets.

Table 5: Effect of different initialization methods on MSCOCO Recall@1 and IN-1K classification.

| Init. Steps | IN-1K Top-1 (%) Acc. | MSCOCO Recall@1 I→T / T→I |
|---|---|---|
| Random Init | 0.1 | 0.02 / 0.01 |
| Kaiming Init (He et al., 2015) | 4.4 | 3.5 / 2.0 |
| Xavier Init (Glorot & Bengio, 2010) | 4.9 | 4.1 / 2.3 |
| **Diag Init (Ours)** | **28.9** | **28.5 / 15.9** |

**Mapping/Retraining Duration.** To determine the optimal number of training epochs for the mapping stage and retraining stage, we conduct 10% compression experiments on the OpenCLIP-ViT-B/16 model. The results are presented in Tab. 4 and the visualization of the weight distribution is presented in A.7. As the mapping stage expands, the distribution of the mapping matrix gradually evolves from an initial diagonal pattern toward a more uniform structure, indicating that the optimization process is progressively searching for an optimal compression mapping. Moreover, as the mapping stage is extended, the performance of the final compressed model is consistently improved. However, we observe that an excessively long mapping stage may lead to performance degradation and introduce unnecessary computational overhead. Thus, we adopt a moderate number of training epochs(5ep) during the mapping stage, which we found providing a good balance between initialization quality and final performance, for both CLIP-Map$_{tiny}$ and CLIP-Map$_{small}$.

## 5 CONCLUSION

In this paper, we propose **CLIP-Map**, a novel *mapping-based* compression framework for CLIP-like multimodal models. CLIP-Map employs learnable transformation matrices to perform layer-wise width compression via matrix multiplication, and leverages learnable linear combinations of parameter blocks across layers to realize depth compression.

To mitigate the overhead of parameter storage and computation introduced by the learnable mappings, we propose a *Full Mapping with Kronecker Factorization* strategy, which significantly reduces the parameter complexity. Furthermore, to address the optimization difficulty, we introduce a *Diagonal Inheritance Initialization* scheme that not only enables weight inheritance from the pretrained model but also improves training stability and convergence, resulting in a well-initialized compressed model.

In addition, we build a complete *mapping-retraining pipeline*, where we incorporate knowledge distillation during retraining to facilitate knowledge transfer from the teacher model. By aligning the output logits of the student and teacher models, our method ensures performance preservation even under aggressive compression settings.

Experimental results on various zero-shot retrieval and zero-shot classification benchmarks demonstrate that CLIP-Map achieves competitive or superior performance compared to existing baselines, while providing significant reductions in model size and training overhead.

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

## A  APPENDIX

In the appendix, we present the detailed design and experiment results of our method.

### A.1  USE OF LLMs

In this paper, we use LLMs to polish writing and to assist with LaTeX typesetting and formatting.

### A.2  ETHICS AND REPRODUCIBILITY STATEMENT

We have reviewed the ICLR Code of Ethics and make sure that our research confirms every respect with the ICLR Code of Ethics. And we list all the concrete settings required in original paper and Appendix and make sure all the results are reproducible.

### A.3  DETAILED ALGORITHM DESIGNING AND MODEL CONFIGURATION

In our setting, both the vision encoder and text encoder of CLIP adopt Transformer-based architectures, which exhibit similar structural forms. For simplicity of notation and without loss of generality, we focus our discussion on the text encoder, as the mapping strategy for the vision encoder follows an analogous formulation.

**Embedding Layer.** For embedding layers, we use learnable matrix $\boldsymbol{F}_{emb}^{out}$ to map and get a smaller embedding layer with reduced embedding dimension.

**Multi-Head Attention Layer.** For the $\ell$-th MHA layer, we have the following components: $\boldsymbol{W}_l^Q, \boldsymbol{W}_l^K, \boldsymbol{W}_l^V, \boldsymbol{W}_l^O$. Instead of assigning distinct $\boldsymbol{F}^{in}$ and $\boldsymbol{F}^{out}$ for each component, we reduce the number of parameters by reusing a shared transformation $\boldsymbol{F}_{emb}^{out}$ across all components. Specially, we set $\boldsymbol{F}_{l,Q}^{in}, \boldsymbol{F}_{l,K}^{in}, \boldsymbol{F}_{l,V}^{in}$ and $\boldsymbol{F}_{l,O}^{out}$ to be $\boldsymbol{F}_{emb}^{out}$, and $\boldsymbol{F}_{l,O}^{in} = \boldsymbol{F}_{l,V}^{out}$.

**Feed-Forward Layer.** For the $\ell$-th FFN layer, we have learnable components $\boldsymbol{W}_l^{fc1}$ and $\boldsymbol{W}_l^{fc2}$. We set $\boldsymbol{F}_{l,fc1}^{in} = \boldsymbol{F}_{emb}^{out}$ and $\boldsymbol{F}_{l,fc2}^{out} = \boldsymbol{F}_{emb}^{out}$.

Table 6: Model configurations for zero-shot image-text retrieval experiments.

| Method | ViT | Vision Encoder | | Text Encoder | | Params (M) | Training Dataset |
|---|---|---|---|---|---|---|---|
| | | Width | Depth | Width | Depth | | |
| SLIP (Mu et al., 2022) | ViT-B/16 | 768 | 12 | 512 | 12 | 86+38 | YFCC-15M |
| CLIP (Wu et al., 2023) | ViT-B/16 | 768 | 12 | 512 | 12 | 86+38 | YFCC-15M |
| CLIP (Radford et al., 2021) | ViT-B/16 | 768 | 12 | 512 | 12 | 86+38 | WIT-400M (Radford et al., 2021) |
| OpenCLIP (Cherti et al., 2023; Ilharco et al., 2021) | ViT-B/16 | 768 | 12 | 512 | 12 | 86+38 | LAION-2B (Schuhmann et al., 2022) |
| MetaCLIP (Xu et al., 2023) | ViT-B/16 | 768 | 12 | 512 | 12 | 86+38 | - |
| CLIP (Radford et al., 2021) | ResNet-50 (He et al., 2016) | - | - | 512 | 12 | 38+38 | WIT-400M |
| CLIP (Radford et al., 2021) | ResNet-101 (He et al., 2016) | - | - | 512 | 12 | 86+38 | WIT-400M |
| *Training data: YFCC15M* | | | | | | | |
| **Compression Ratio: 1.0%** | | | | | | | |
| TinyCLIP (Wu et al., 2023) | ViT-B/16 | 128 | 4 | 128 | 2 | 0.8+0.3 | YFCC15M |
| † TinyCLIP (Wu et al., 2023) (3×25ep) | ViT-B/16 | 128 | 4 | 128 | 2 | 0.8+0.3 | YFCC15M |
| **CLIP-Map$_{tiny}$ (Ours)** | ViT-B/16 | 512 | 12 | 512 | 6 | 0.8+0.3 | YFCC15M |
| **Compression Ratio: 10.0%** | | | | | | | |
| TinyCLIP (Wu et al., 2023) | ViT-B/16 | 256 | 10 | 256 | 3 | 8+3 | YFCC15M |
| † TinyCLIP (Wu et al., 2023) (2×25ep) | ViT-B/16 | 256 | 10 | 256 | 3 | 8+3 | YFCC15M |
| **CLIP-Map$_{small}$ (Ours)** | ViT-B/16 | 256 | 10 | 256 | 3 | 8+3 | YFCC15M |
| **CLIP-Map$_{small}$ (Ours, Meta-CLIP)** | ViT-B/16 | 256 | 10 | 256 | 3 | 8+3 | YFCC15M |
| **Compression Ratio: 50.0%** | | | | | | | |
| TinyCLIP (Wu et al., 2023) | ViT-B/16 | 512 | 24 | 768 | 6 | 39+19 | YFCC15M |
| **CLIP-Map$_{base}$ (Ours)** | ViT-B/16 | 512 | 12 | 512 | 6 | 39+19 | YFCC15M |
| **CLIP-Map$_{base}$ (Ours, Meta-CLIP)** | ViT-B/16 | 512 | 12 | 512 | 6 | 39+19 | YFCC15M |
| **CLIP-Map$_{base}$ (Ours, ResNet-50, wo Retraining)** | ResNet-19M | - | - | 512 | 6 | 19+19 | YFCC15M |

**Training Loss.** In the mapping stage, we train the mapping parameters using standard CLIP task loss as in Eq. 12.

**Detailed Model Configuration.** Detailed model architectures are provided in Tab. 6 to complement the results in Tab. 1.

## A.4 COMPARISON WITH OTHER METHOD

Table 7: Comparison of image–text retrieval performance.

| Method | Dataset | Params (img + text) | I2T@1 | I2T@5 | I2T@10 | T2I@1 | T2I@5 | T2I@10 |
|---|---|---|---|---|---|---|---|---|
| UPop (Shi et al., 2023a) | MSCOCO | 280.2M | 56.1 | 82.4 | 90.2 | 41.1 | 71.0 | 81.4 |
| EfficientVLM (Wang et al., 2022b) | CC3M | 152M + 42M | 46.6 | 71.7 | 81.3 | 35.9 | 61.6 | 71.8 |
| DynaCLIP (Hou et al., 2020) | CC3M | 86M + 42M | 51.3 | 75.5 | 84.6 | 35.8 | 61.8 | 72.6 |
| CLIP-Map-base (**Ours**) | YFCC15M | 39M + 19M | 55.1 | 78.8 | 86.5 | 37.9 | 63.8 | 74.1 |
| CLIP-Map-small (**Ours**) | YFCC15M | 8M + 3M | 38.4 | 64.6 | 74.4 | 24.3 | 48.4 | 59.9 |

To further validate the generalization capability of our approach, we compare its performance with other compression methods. We have surveyed several other VLM compression methods, including UPop (Shi et al., 2023a) and EfficientVLM (Wang et al., 2022b) as in Tab.7. We find that these methods don't report experiments at higher compression ratios, and the compressed models remain relatively large (over 100M parameters), even though they achieve high performance. However, reproducing these methods would require considerable time and implementation effort. Therefore, due to time constraints, we don't adapt these methods to our experimental setting.

To establish baselines, we adopt the existing retrieval results on the MSCOCO dataset reported in the respective papers of the compared methods and use them for comparison with our proposed approach. It should be noted that DynaCLIP is the result of transferring the DynaBERT (Hou et al., 2020) method to the CLIP framework, and the reported performance is sourced from MoPE-CLIP (Lin et al., 2024).

## A.5 DETAILED TRAINING SETTINGS

Tab. 8 and Tab. 9 display the detailed training hyperparameters to get three scales of CLIP-Map models on YFCC15M dataset. Considering the smaller model capacity and weaker representational ability of CLIP-Map$_{small}$ and CLIP-Map$_{tiny}$, we employ a longer mapping stage. For the larger CLIP-Map$_{base}$ variant, a shorter mapping stage proves adequate. All models are trained on 32 Nvidia H800 and it takes around 15 hours to finish the mapping-retraining pipeline for these three variant models each.

Table 8: Hyperparameters used in the Mapping Stage for different CLIP-Map model sizes.

| Hyperparameter | CLIP-Map$_{tiny}$ | CLIP-Map$_{small}$ | CLIP-Map$_{base}$ |
|---|---|---|---|
| Batchsize | 4096 | 4096 | 4096 |
| Epochs | 5 | 5 | 0.28(1000steps) |
| Distillation | False | False | False |
| Optimizer | AdamW | AdamW | AdamW |
| Optimizer Momentum | $\beta_1 = 0.9, \beta_2 = 0.98$ | $\beta_1 = 0.9, \beta_2 = 0.98$ | $\beta_1 = 0.9, \beta_2 = 0.98$ |
| Base Learning Rate | $8 \times 10^{-4}$ | $8 \times 10^{-4}$ | $2 \times 10^{-3}$ |
| Weight Decay | 0.2 | 0.2 | 0.2 |
| Learning Rate Schedule | Cosine decay | Cosine decay | Cosine decay |
| Warmup (steps) | 2000 | 2000 | 1000 |
| Gradient Clipping Norm | 5 | 5 | 5 |
| Reciprocal of Temperature | 50 | 50 | 50 |
| Image Resolution | $224 \times 224$ | $224 \times 224$ | $224 \times 224$ |
| Image Augmentation | RandomResizedCrop | RandomResizedCrop | RandomResizedCrop |
| Tokenizer | Byte Pair Encoding | Byte Pair Encoding | Byte Pair Encoding |
| Vocabulary Size | 49,408 | 49,408 | 49,408 |
| Max Sequence Length | 77 | 77 | 77 |

Table 9: Hyperparameters used in the Retraining Stage for different CLIP-Map model sizes.

| Hyperparameter | CLIP-Map$_{tiny}$ | CLIP-Map$_{small}$ | CLIP-Map$_{base}$ |
|---|---|---|---|
| Batchsize | 4096 | 4096 | 8192 |
| Epochs | 25 | 25 | 25 |
| Optimizer | AdamW | AdamW | AdamW |
| Optimizer Momentum | $\beta_1 = 0.9, \beta_2 = 0.98$ | $\beta_1 = 0.9, \beta_2 = 0.98$ | $\beta_1 = 0.9, \beta_2 = 0.98$ |
| Base Learning Rate | $1 \times 10^{-4}$ | $1 \times 10^{-4}$ | $3 \times 10^{-4}$ |
| Weight Decay | 0.2 | 0.2 | $1 \times 10^{-4}$ |
| Learning Rate Schedule | Cosine decay | Cosine decay | Cosine decay |
| Warmup (steps) | 2000 | 2000 | 2000 |
| Gradient Clipping Norm | 5 | 5 | 5 |
| Reciprocal of Temperature | 50 | 50 | 50 |
| Image Resolution | $224 \times 224$ | $224 \times 224$ | $224 \times 224$ |
| Image Augmentation | RandomResizedCrop | RandomResizedCrop | RandomResizedCrop |
| Tokenizer | Byte Pair Encoding | Byte Pair Encoding | Byte Pair Encoding |
| Vocabulary Size | 49,408 | 49,408 | 49,408 |
| Max Sequence Length | 77 | 77 | 77 |

Table 10: Performance comparison under different $\lambda$ values.

| $\lambda$ | 0.1 | 0.2 | 0.3 | 0.4 | 0.5 | 0.6 | 0.7 | 0.8 | 0.9 | 1.0 |
|---|---|---|---|---|---|---|---|---|---|---|
| MSCOCO TR@1 | 33.1 | 37.2 | 40.1 | 42.2 | 43.8 | 45.6 | 48.1 | 49.4 | 50.2 | **51.1** |
| MSCOCO IR@1 | 23.4 | 26.2 | 27.7 | 29.0 | 30.6 | 31.6 | 33.2 | 34.0 | **34.7** | **34.7** |

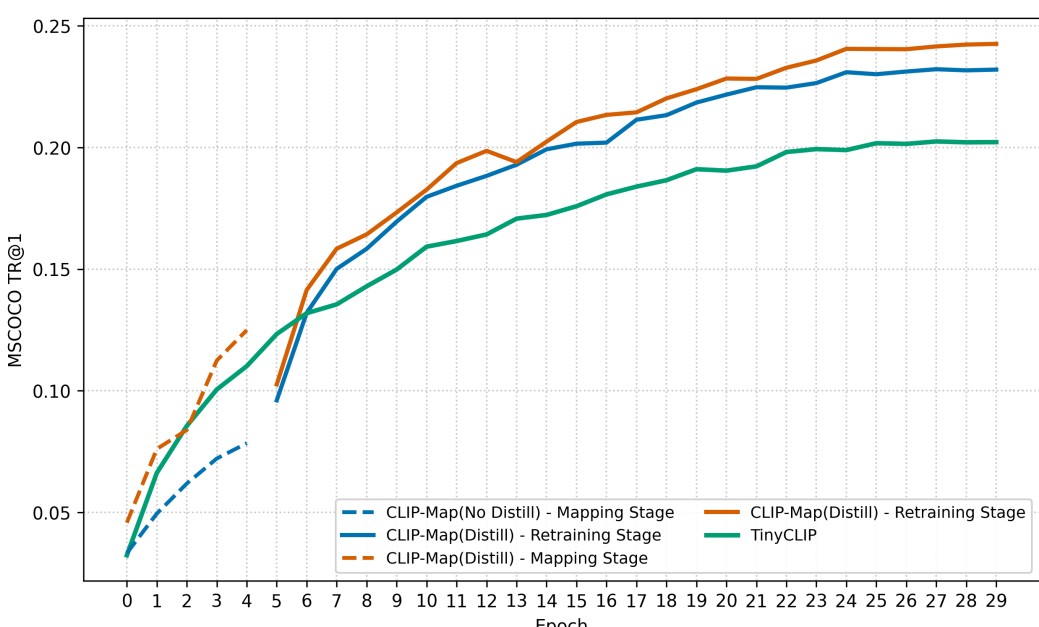

Figure 4: Evolution of TR@1 on the MSCOCO test set using TinyCLIP and CLIP-Map(w and w/o distillation during mapping stage).

## A.6 TRAINING AND INFERENCE SPEED-UP

Table 11: Wall-clock training time and epochs for different model variants.

| Model | Wall-Clock Time (Multi-stages) | Epochs |
|---|---|---|
| TinyCLIP$_{base}$ | 14h43m | 25ep (25ep) |
| **CLIP-Map$_{base}$** | **14h30m** (6m + 14h23m) | 25ep (1000steps + 25ep) |
| TinyCLIP$_{small}$ | 32h36m (14h43m + 17h53m) | 50ep (25ep + 25ep) |
| **CLIP-Map$_{small}$** | **22h10m** (4h50m + 17h19m) | 25ep (5ep + 25ep) |
| TinyCLIP$_{tiny}$ | 49h33m (14h43m + 17h53m + 16h57m) | 75ep (25ep + 25ep + 25ep) |
| **CLIP-Map$_{tiny}$** | **21h50m** (4h37m + 17h11m) | 25ep (5ep + 25ep) |

We record the wall-clock time required for training different model variants, with the results presented in Tab. 11. Meanwhile, we also visualize the evolution of text-to-image retrieval performance (TR@1) on the MSCOCO test set during training, as shown in Fig. 4. Specifically, we compare our CLIP-Map method against TinyCLIP under the same compression setting (compressing OpenCLIP-ViT-B/16 to the *small* variant). Moreover, we investigate whether incorporating distillation during the mapping stage can further enhance performance. The results demonstrate that our method achieves faster convergence and consistently outperforms the baseline throughout training. It is worth noting that although incorporating distillation during the mapping stage can lead to a slight performance gain, we observe a drop in accuracy at the beginning of the retraining stage. We argue that such an instability may hinder subsequent optimization. Therefore, in our experiments, we avoid using distillation during the mapping stage in all subsequent experiments.

Table 12: Comparison of computational cost(GFLOPs/pair) and throughput(Pairs/s) across different model variants.

| Model | GFLOPs (img + text) | Throughput (Pairs/s) |
|---|---|---|
| ViT-B/16 | $17.56 + 2.98$ | 1041 pairs/s |
| CLIP-Map$_{base}$ / TinyCLIP$_{base}$ | $7.99 + 1.49$ | 1654 pairs/s ($\times$**1.6**) |
| CLIP-Map$_{small}$ / TinyCLIP$_{small}$ | $1.79 + 0.19$ | 2466 pairs/s ($\times$**2.4**) |
| CLIP-Map$_{tiny}$ / TinyCLIP$_{tiny}$ | $0.21 + 0.03$ | 3847 pairs/s ($\times$**3.8**) |

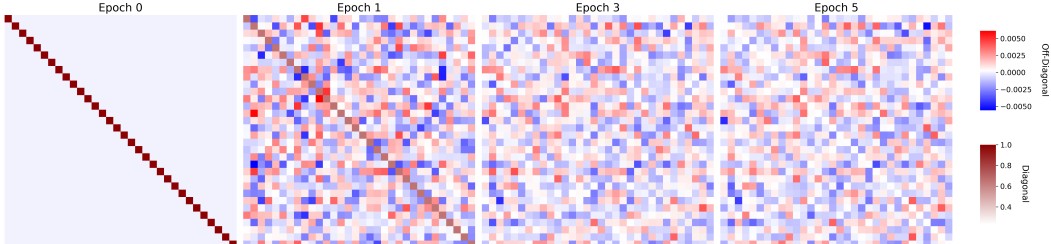

Figure 5: Changes of a mapping matrix in CLIP-Map$_{small}$ mapping stage. Due to the significant scale difference between diagonal and off-diagonal weights, we adopt two separate color scales to enhance the visualization quality.

The enhancement in the inference efficiency of our model is measured using the throughput of image–text pairs and the GFLOPs cost per image-text pair. All metrics are tested on single H800 GPU, at a batch size of 32 with an image resolution of 224×224, shown in Tab. 12. Because our compressed model shares the same parameter count and architecture as its TinyCLIP counterpart of equivalent size, the FLOPs and throughput measurements are identical for both.

## A.7 VISUALIZATION

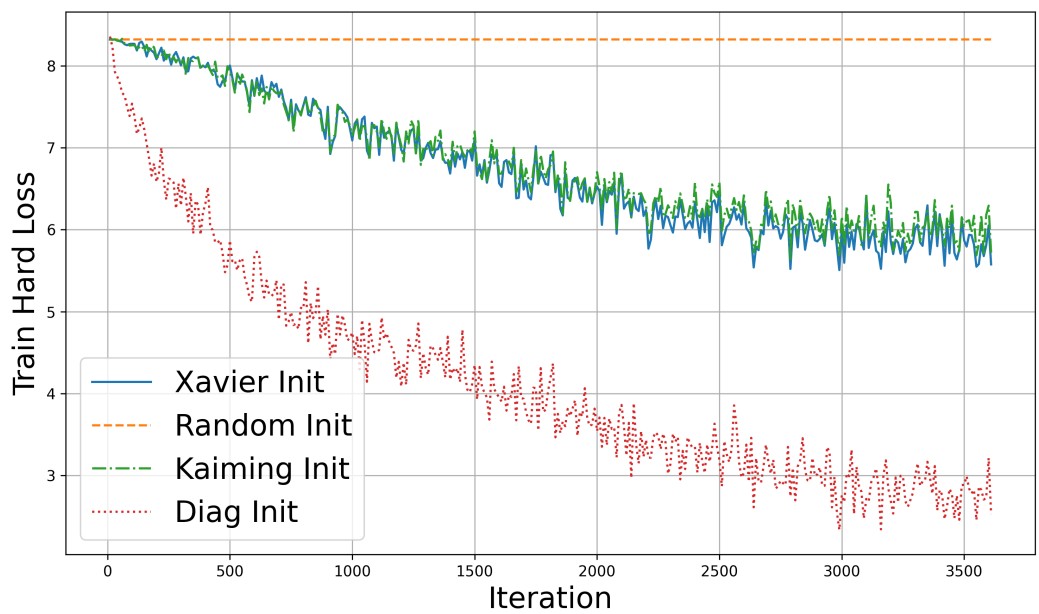

Figure 6: Loss curves under different initialization methods.

To better understand the mapping stage and the effect of different initialization methods, we visualize the changes in the mapping matrices across different epochs during the mapping stage of CLIP-Map$_\text{small}$ training in Fig. 5 and loss curves under different initialization methods during mapping stage in Fig. 6. For clarity, we only present a subset of the parameter matrix.

## A.8 Ablation on Training Loss

To efficiently identify the optimal distillation coefficient $\lambda$, we conduct a series of experiments on the small-scale CC3M dataset. In this setting, we compress the OpenCLIP-ViT-B/16 model into CLIP-Map$_\text{small}$, and evaluate performance on MSCOCO test set under various $\lambda$ values. The results are reported in Tab. 10. As the value of $\lambda$ increases, the distillation loss becomes more dominant during training, leading to consistent performance improvements. Notably, when $\lambda = 1.0$, i.e., the total loss is fully composed of the distillation objective ($\mathcal{L}_\text{total} = \mathcal{L}_\text{distill}$), the model achieves its best performance. Therefore, we adopt $\lambda = 1.0$ as the default setting in all subsequent experiments.

## A.9 Limitations and Future Works

In this paper, we propose a mapping-based compression method termed CLIP-Map, using learnable matrices to mapping the original model into a smaller one. Our CLIP-Map outperforms baseline methods under equivalent compression settings, demonstrating both effectiveness and efficiency. However, our study is constrained by limited computational resources, which prevented us from evaluating the method on larger-scale datasets.

In future work, we plan to extend our approach to larger training datasets such as LAION-2B (Schuhmann et al., 2022) to further validate its effectiveness and generalizability.

## A.10 Broader Impacts

This work focuses on model compression techniques for CLIP, with the aim of enabling efficient deployment on resource-constrained devices. Since CLIP primarily performs vision-language understanding rather than generation, it is less likely to introduce negative societal impacts. To the best of our knowledge, our paper does not pose any negative societal impact.

