# OpenReview forum: "CLIP-Map: Structured Matrix Mapping for Parameter-Efficient CLIP Compression"
_ICLR.cc/2026/Conference — Submitted to ICLR 2026_

### Official Review · Reviewer_24zq · 2025-10-24

**Soundness:** 3
**Presentation:** 2
**Contribution:** 3
**Rating:** 4
**Confidence:** 4

**Summary:**

This paper proposes a new framework CLIP-Map, which uses learnable matrices to map and combine pre-trained weights to retain more original weight information; it also addresses optimization challenges by reducing distribution shifting. Experiments show CLIP-Map outperforms select-based frameworks across compression ratios, with notable gains under high compression.

**Strengths:**

1. It introduces Kronecker product to reduce the complexity.

2. The testing datasets including general and specific domain as shown in Table 2 is adequate for the experiments.

3. It demonstrates three types of CLIP-Map(base, small and tiny), which can meet different requirements.

**Weaknesses:**

1. For depth compression, it assumes the new layer can be expressed as a linear function of the old layer but it may not be true. Its primary goal is to preserve a portion of the original weights, which is already demonstrated by diagonal initialization. Can we also construct other Kronecker product matrices with a block product of 1 to preserve some weights?

2. The performance drops dramatically especially on Stanford Cars(from 88.5 to 50.8) and MNIST(from 65.8 to 13.0).

3. The results of other methods like quantization and distillation shall be compared.

**Questions:**

This paper essentially performs selective initialization on weights, which can reduce training overhead and thereby improve efficiency. The question then arises: whether there exist more efficient initialization methods, such as Weakness 1.

---

> ### Author Response · Authors · 2025-11-23
>
> **R Q1 && W1: For depth compression, it assumes the new layer can be expressed as a linear function of the old layer but it may not be true. Its primary goal is to preserve a portion of the original weights, which is already demonstrated by diagonal initialization. Can we also construct other Kronecker product matrices with a block product of 1 to preserve some weights? Whether there exist more efficient initialization methods**
>
> We also experimented with other Kronecker product matrices whose block-wise product equals 1, for example initializing the corresponding positions of the mapping matrices to 1 on alternating rows. However, we found that the final performance was very similar to that obtained with our diagonal initialization.
>
> **R W2:  The performance drops dramatically especially on Stanford Cars(from 88.5 to 50.8) and MNIST(from 65.8 to 13.0).**
>
> We believe that the significant performance drop on these two datasets is mainly due to their inherent characteristics. Both the Stanford Cars and MNIST datasets are highly sensitive to model compression, and we observed that the baseline methods we compared against also experienced substantial degradation after compression on these datasets.
>
> **R W3: The results of other methods like quantization and distillation shall be compared.**
>
> We compare our method with other CLIP-based approaches, including PromptKD[1], CLIP-KD[2], and MaskedCLIP[3]. The results are shown in the tables below.
>
> | Method             | Params(M) | Caltech | Oxford Pets | Stanford Cars | Flowers 102 | Food101 | FGVC Aircraft | SUN397 | DTD   | EuroSAT | UCF101 | Avg.  |
> | ------------------ | --------- | ------- | ----------- | ------------- | ----------- | ------- | ------------- | ------ | ----- | ------- | ------ | ----- |
> | PromptKD(ViT-B/16) | 124       | 93.61   | 91.59       | 73.93         | 75.33       | 88.84   | 26.24         | 68.57  | 55.08 | 63.74   | 76.39  | 71.33 |
> | MaskCLIP(ViT-B/16) | 124       | 72.0    | 34.3        | 6.7           | 57.0        | 64.9    | -             | -      | 27.9  | 44.0    | -      | 43.83 |
> | CLIP-Map(Base)     | 58        | 81.4    | 83.1        | 50.8          | 70.2        | 82.7    | 22.2          | 69.2   | 48.5  | 52.6    | -      | 62.30 |
>
> | Method             | Params(M) | IN(Zeroshot) acc@1 | MSCOCO I2T@1 | MSCOCO I2T@5 | MSCOCO I2T@10 | MSCOCO T2I@1 | MSCOCO T2I@5 | MSCOCO T2I@10 | Flicke30k I2T@1 | Flicke30k I2T@5 | Flicke30k I2T@10 | Flicke30k T2I@1 | Flicke30k T2I@5 | Flicke30k T2I@10 |
> | ------------------ | --------- | ------------------ | ------------ | ------------ | ------------- | ------------ | ------------ | ------------- | --------------- | --------------- | ---------------- | --------------- | --------------- | ---------------- |
> | CLIP-KD(ViT-T/16)  | 5.72      | 34.9               | 23.1         | -            | -             | 22.6         | -            | -             | 52.3            | -               | -                | 52.4            | -               | -                |
> | MaskCLIP(ViT-B/16) | 124       | 44.5               | 41.4         | 67.9         | 77.5          | 25.5         | 49.7         | 61.3          | 70.1            | 90.3            | 95.3             | 45.6            | 73.4            | 82.1             |
> | CLIP-Map(small)    | 11        | 42.7               | 55.1         | 78.8         | 86.5          | 37.9         | 63.8         | 74.1          | 81.9            | 96.2            | 98.8             | 67.6            | 89.0            | 94.0             |
> | CLIP-Map(base)     | 58        | 63.7               | 38.4         | 64.6         | 74.4          | 24.3         | 48.4         | 59.9          | 66.0            | 88.5            | 92.6             | 46.9            | 73.9            | 82.6             |
>
> The results demonstrate that, compared with MaskCLIP and CLIP-KD, our method achieves superior performance while using fewer parameters. In contrast, although our approach is less competitive than PromptKD, which employs prompt tuning, this is largely because PromptKD does not compress the model and introduces additional knowledge through prompt tuning. Nevertheless, considering both overall parameter count and performance, our model attains a very favorable trade-off.
>
>
> [1] Li Z, Li X, Fu X, et al. Promptkd: Unsupervised prompt distillation for vision-language models[C]//Proceedings of the IEEE/CVF Conference on Computer Vision and Pattern Recognition. 2024: 26617-26626.
>
> [2] Yang C, An Z, Huang L, et al. Clip-kd: An empirical study of clip model distillation[C]//Proceedings of the IEEE/CVF Conference on Computer Vision and Pattern Recognition. 2024: 15952-15962.
>
> [3] Dong X, Bao J, Zheng Y, et al. Maskclip: Masked self-distillation advances contrastive language-image pretraining[C]//Proceedings of the IEEE/CVF conference on computer vision and pattern recognition. 2023: 10995-11005.

---

> > ### Author Response · Authors · 2025-11-25
> >
> > We hope this message finds you well. As the deadline is approaching, we would greatly appreciate it if you could kindly provide your feedback at your earliest convenience. Thank you very much for your time and consideration.

---

> > ### Comment · Reviewer_24zq · 2025-11-27
> > **Response to authors**
> >
> > Thank you for your detailed responses to the raised weaknesses. However, after carefully reviewing your replies and the referenced works (PromptKD [1], CLIP-KD [2], MaskedCLIP [3]), I find that the core concerns are not adequately addressed, which will lead to a lower score. Here are the specific reasons:
> >
> > ### 1. Unresolved core assumption of linear mapping for depth compression
> > Your response only mentions that other Kronecker product matrices (e.g., alternating rows with 1s) yield similar performance to diagonal initialization, but completely avoids addressing the key question: *whether the assumption that "new layers can be expressed as a linear function of old layers" is valid*. The diagonal initialization merely preserves partial weight information, but it does not verify or justify the rationality of the linear mapping hypothesis itself. If this core assumption does not hold for CLIP’s feature learning mechanism (e.g., non-linear interactions in contrastive pretraining), the entire mapping-based compression framework lacks a solid theoretical foundation—this critical gap remains unaddressed.
> >
> > ### 2. Lack of empirical evidence to support "dataset sensitivity" for performance drops
> > You attribute the dramatic performance degradation on Stanford Cars (88.5→50.8) and MNIST (65.8→13.0) to dataset sensitivity, but fail to provide **specific degradation data of baseline methods on these two datasets** (e.g., MaskedCLIP, CLIP-KD’s performance before/after compression on Stanford Cars/MNIST). Without such comparative data, the claim of "baseline methods also experiencing substantial degradation" is unsubstantiated. Moreover, even if datasets are sensitive to compression, a robust compression method should mitigate such degradation rather than merely attributing it to data characteristics—this response does not address how to improve the model’s adaptability to sensitive datasets.
> >
> > ### 3. Incomplete comparison with target compression methods
> > The original concern was to compare with **quantization and distillation-based compression methods** (two mainstream model compression paradigms), but your response only adds comparisons with PromptKD, CLIP-KD, and MaskedCLIP. Among these, PromptKD does not involve model compression (it uses prompt tuning to introduce additional knowledge), and CLIP-KD/MaskedCLIP belong to distillation/self-distillation frameworks similar to your method—none of them are quantization-based methods. This means the comparison still lacks coverage of a key compression paradigm (quantization), making it impossible to fully demonstrate CLIP-Map’s competitiveness across different compression approaches.
> >
> > In summary, the core assumptions, empirical support, and comparative scope of the raised weaknesses remain unaddressed. These gaps affect the credibility and comprehensiveness of the work, so I have to reduce the score accordingly.

---

> > > ### Author Response · Authors · 2025-11-30
> > >
> > > **R (1)** The assumption that “new layers can be expressed as a linear function of old layers” is inspired by model-growth methods such as Net2Net[1]. In these approaches, a larger network is constructed by preserving the layers of a smaller pretrained model and inserting identity layers between them as the initialization for the expanded architecture. After subsequent training, such models often achieve strong performance. This model-growth mechanism can be viewed as implicitly assuming that newly added layers can be represented as a linear transformation or linear combination of the original layers. Motivated by this observation, we adopt a similar idea and parameterize the new layers as linear combinations of the corresponding layers in the original model.
> > >
> > > **R (2)** We believe that the low image resolutions of the Stanford Cars and MNIST datasets make them considerably more sensitive to model compression. As shown in the tables included in our manuscript, MaskedCLIP also experiences substantial performance degradation on these two datasets. This consistent decline across different methods suggests that the sensitivity primarily arises from the dataset characteristics rather than from our method alone.
> > >
> > > **R (3)** Our method is specifically designed to target pruning-based distillation frameworks, such as TinyCLIP. Since existing frameworks in this line of work, including TinyCLIP and MaskedCLIP, do not incorporate comparisons with quantization-based compression methods, we followed the same evaluation protocol and did not include quantization approaches in our comparison.
> > >
> > > [1] Chen T, Goodfellow I, Shlens J. Net2net: Accelerating learning via knowledge transfer[J]. arXiv preprint arXiv:1511.05641, 2015.

---

> > > > ### Author Response · Authors · 2025-12-02
> > > >
> > > > As the deadline is approaching, we are keen to know whether our previous response has fully addressed your concerns.

---

> ### Author Response · Authors · 2025-11-23
>
> We hope that our answers have thoroughly addressed your questions, and we kindly look forward to your response.

---

### Official Review · Reviewer_TpdV · 2025-10-27

**Soundness:** 3
**Presentation:** 3
**Contribution:** 2
**Rating:** 6
**Confidence:** 3

**Summary:**

This paper proposes the CLIP-Map model compression framework. Unlike existing approaches, CLIP-Map addresses the optimisation challenge of mapping matrices by leveraging Kronecker factorisation and learnable inter-layer linear combinations. The authors also introduce a diagonal inheritance initialisation technique. However, this paper also exhibits several significant limitations.

**Strengths:**

This paper addresses a practical and timely issue: large multimodal models incur substantial computational and storage overhead, which is critical for resource-constrained devices. Furthermore, the core approach of replacing selection and discard with mapping and combining pre-trained weights is reasonably justified. This method can preserve the original model's information, demonstrating particularly significant effects at high compression rates, whereas comparable methods may result in critical information loss. The authors' experiments further demonstrate that CLIP-Map achieves substantial performance gains under extreme compression ratios over the TinyCLIP baseline model, thereby validating its advantages.

**Weaknesses:**

1) This paper's core idea is that mapping preserves more information than selection currently remains at an intuitive level, lacking rigorous theoretical support. For instance, the authors fail to analyse from a mathematical perspective why this mapping-based structure can more effectively maintain the knowledge fidelity of pre-trained models. 2) The experiments lack ablation studies. For instance, the paper provides no ablation experiments to distinguish the respective contributions of width and depth compression.

**Questions:**

The current experiments appear to omit a key ablation study: how much do these two components contribute to the model's performance?

---

> ### Author Response · Authors · 2025-11-25
>
> **R W1: This paper's core idea is that mapping preserves more information than selection currently remains at an intuitive level, lacking rigorous theoretical support. For instance, the authors fail to analyse from a mathematical perspective why this mapping-based structure can more effectively maintain the knowledge fidelity of pre-trained models.**
>
> Although there is no strong mathematical proof that the mapping-based approach is strictly better than the selection-based approach, this conclusion is mainly supported by extensive prior work and our empirical results. In the **Experiments** section, we provide a detailed comparison between mapping-based and selection-based methods on multiple benchmarks. The results consistently demonstrate that the mapping-based approach is more effective, especially under low compression ratios.
>
>
>
> **R W2: The experiments lack ablation studies. For instance, the paper provides no ablation experiments to distinguish the respective contributions of width and depth compression.**
>
> We conducted ablation studies where depth compression and width compression were applied separately. We first applied width compression, followed by depth compression. The results clearly show that depth compression, which removes a large number of layers, has a much larger impact on the overall accuracy of the model. In contrast, width compression reduces fewer parameters and therefore causes a significantly smaller accuracy drop.

---

> > ### Author Response · Authors · 2025-11-27
> >
> > We hope this message finds you well. As the deadline is approaching, we would greatly appreciate it if you could kindly provide your feedback at your earliest convenience. Thank you very much for your time and consideration.

---

### Official Review · Reviewer_V2MN · 2025-10-28

**Soundness:** 3
**Presentation:** 3
**Contribution:** 3
**Rating:** 4
**Confidence:** 3

**Summary:**

This paper proposes a mapping-based compression method. It utilizes Kronecker factorization to map pre-trained weights to a lower dimension to reduce model size. To train the mapping matrix, this paper also introduces Diagonal Inheritance Initialization to alleviate distribution shift issues and stabilize training. The effectiveness and efficiency of this method are demonstrated in extreme compression scenarios with high compression rates.

**Strengths:**

The methods proposed in this paper are technically sound, and both contributions offer relevant improvements for clip compression. Moreover, the experiments are comprehensive, with thorough exploration conducted on both classification and retrieval tasks. This task is also quite important.

**Weaknesses:**

Can the compressed CLIP model be widely applied to downstream tasks? The experiments in this paper only evaluated retrieval and classification tasks, without assessing other tasks such as generation or some comprehension tasks. If the model can only improve performance in classification and retrieval but not be applicable to these other tasks, its applicability in a broader range of fields would be significantly reduced. It would be even better if comparisons with other compressed models on other tasks could be provided.

Diagonal initialization is one of the main contributions of this paper, which is also experimentally validated. However, the improvement compared to other initialization methods is quite significant. The diagonal initialization method has been mentioned in other works (e.g., "On the Parameterization and Initialization of Diagonal State Space Models"), and it is not a novel algorithm. Therefore, its innovation seems limited.

**Questions:**

see above .The depth compression part in Figure 3 could be made clearer.

---

> ### Comment · Reviewer_V2MN · 2025-11-28
>
> after read the rebuttal, I will raise my score to 6, marginally above the acceptance threshold

---

> ### Author Response · Authors · 2025-11-30
>
> **R W1: Can the compressed CLIP model be widely applied to downstream tasks? The experiments in this paper only evaluated retrieval and classification tasks, without assessing other tasks such as generation or some comprehension tasks. If the model can only improve performance in classification and retrieval but not be applicable to these other tasks, its applicability in a broader range of fields would be significantly reduced. It would be even better if comparisons with other compressed models on other tasks could be provided.**
>
> We sincerely apologize that we were unable to evaluate our compressed models on other downstream tasks. However, we believe that our compression method may not yield significant advantages in these scenarios. This is because our method applies a relatively high compression ratio, which is primarily designed for retrieval and classification tasks. For tasks such as visual understanding (e.g., LLaVA[1]) and generation (e.g., Stable Diffusion[2]), such a compact model is likely to result in a substantial performance degradation.
>
> **R W2: Diagonal initialization is one of the main contributions of this paper, which is also experimentally validated. However, the improvement compared to other initialization methods is quite significant. The diagonal initialization method has been mentioned in other works (e.g., "On the Parameterization and Initialization of Diagonal State Space Models"), and it is not a novel algorithm. Therefore, its innovation seems limited.**
>
> We believe that our primary novelty lies in reformulating model parameter compression as an optimization problem through the use of a mapping matrix. The Kronecker decomposition and the diagonal initialization are employed to address two key challenges: the large number of parameters in the mapping matrix and the instability during training, respectively. In the paper you mentioned, the diagonal initialization is used to address the gradients scaling exponentially in the sequence length problem, and by using diagonal initialization, SSMs can better memorize history input. In our paper, diagonal initialization is used to memorize the "state of teacher model" to solve the unstable training problem.

---

### Official Review · Reviewer_dmgx · 2025-11-01

**Soundness:** 3
**Presentation:** 3
**Contribution:** 3
**Rating:** 4
**Confidence:** 3

**Summary:**

In this paper, the authors studied the problem of compressing CLIP models efficiently while preserving performance. They proposed CLIP-Map, a mapping-based CLIP compression framework that uses learnable structured matrices to map and combine pretrained weights instead of simply selecting subsets of weights. The method leverages full-mapping with Kronecker factorization and a Diagonal Inheritance Initialization strategy to preserve original weight distributions. The compressed model is first learned through mapping while freezing the original CLIP, followed by a knowledge-distillation retraining stage using the original CLIP as a teacher. Experiments show that CLIP-Map outperforms selection-based compression approaches across various compression ratios, with especially strong performance under high compression settings

**Strengths:**

- The paper is well-organized and easy to follow.
- The proposed method is effective, and the use of structured matrix mapping is novel in the context of CLIP compression.
- The experimental results are comprehensive and convincing.

**Weaknesses:**

- This paper greatly violates ICLR's formatting guidelines (see suggestions below). The paper should not be accepted in its current form.
- Equation (10) is not rigorous. The kronecker product of $F^{in}$ and $F^{out}$ initialized with Equation (9) is not an identity matrix (the diagonal entries are not all 1 and there are non-zero off-diagonal entries)

**Questions:**

- Questions
  - What does the kronecker product of $F^{in}$ and $F^{out}$ look like after the first stage?
  - In your experiments, is it possible to directly train the mapping matrices without decomposing them into kronecker products? If so, how does the performance compare with CLIP-Map?
- Suggestions
  - The font size in the tables is too small.
  - The margin between headings and texts is small as well.
  - I suggest the authors provide the average performance across 21 datasets in Table 2.
  - The captions of tables should be placed above the tables.
- Typos
  - Line 70: using -> use
  - Line 149: TinyCLIPWu et al. (2023) -> TinyCLIP (Wu et al., 2023)
  - Line 150: other -> another
  - Line 278: leading -> leading to
  - Figure 3: the subscripts of $F^{in}$ and $F^{out}$ seems incorrect (e.g., there are three $F_{2, 3}^{out}$ in $F^{out}$)
  - Line 420: The bold texts are a paragraph itself, which is inconsistent with previous bold texts that are part of a paragraph.
  - Line 466: to provide -> provides

---

> ### Author Response · Authors · 2025-11-17
>
> **R W1: This paper greatly violates ICLR's formatting guidelines (see suggestions below). The paper should not be accepted in its current form.**
>
> First, we sincerely apologize for the formatting errors in our paper. Thank you for pointing out the errors in our article. We will learn from this experience and ensure that such issues are corrected in the future. We are truly sorry for the inconvenience caused, and we have revised the paper’s formatting as per your valuable suggestions..
>
> **R W2: Equation (10) is not rigorous. The kronecker product of $F^{in}$ and $F^{out}$ initialized with Equation (9) is not an identity matrix (the diagonal entries are not all 1 and there are non-zero off-diagonal entries)**
>
> You are right that the product of $F^{in}$ and $F^{out}$ are indeed not identity matrices, and the description around Equation (9) is not precise. Our original intention in using this formulation was to indicate that we partially preserve the original model parameters via a transformation that behaves similarly to an identity mapping.
>
> For example, for a weight matrix in the original model $W \in \mathbb{R}^{D_1 \times D_1}$, we can construct a mapping matrix of size $\mathbb{R}^{D_1^2 \times D_2^2}$, where the entry is 1 only when the row index i equals the column index j, and 0 otherwise. If we flatten the original weight W into a vector $V \in \mathbb{R}^{D_1^2}$, then this mapping matrix can be applied through standard matrix–vector multiplication to realize the compression of that layer while approximately preserving the original parameters.
>
> Therefore, in Equation (10), our intention was to convey that the keonecker product of $F^{in}$ and $F^{out}$ act approximately like an identity mapping on the original parameters(just like the example above), rather than being strict identity matrices in the algebraic sense. We acknowledge that our wording was misleading and may have caused confusion, and we will revise the text to clarify this point.
>
> We will update the descriptions around Equations (9) and (10) to provide a more rigorous and accurate explanation.
>
> **R Q1: What does the kronecker product of $F^{in}$  and  $F^{out}$ look like after the first stage?**
>
> After the first-stage optimization, the distributions of $F^{in}$ and $F^{out}$ are shown in Appendix A.8, Figure 6. At initialization, only the elements on the main diagonal are set to 1 and all off-diagonal entries are 0. During optimization, the off-diagonal entries are slightly updated and take small non-zero values (approximately in the range $-0.005 \sim 0.005$), while the main-diagonal entries retain relatively large magnitudes (around $0.4 \sim 1.0$).
>
> **R Q2: In your experiments, is it possible to directly train the mapping matrices without decomposing them into kronecker products? If so, how does the performance compare with CLIP-Map?**
>
> Note that if we do not use the Kronecker-factorized form and directly parameterize the mapping matrix, it would contain D_1 D_1 D_2 D_2 = D_1^2 D_2^2 parameters. Such a parameter explosion would make the model much more expensive to train, both in terms of memory and computation, which is precisely what our Kronecker design aims to avoid.

---

> > ### Author Response · Authors · 2025-11-25
> >
> > We hope this message finds you well. As the deadline is approaching, we would greatly appreciate it if you could kindly provide your feedback at your earliest convenience. Thank you very much for your time and consideration.

---

> ### Author Response · Authors · 2025-11-23
>
> We hope that our answers have thoroughly addressed your questions, and we kindly look forward to your response.

---

### Author Response · Authors · 2025-11-27
**PLEASE ENGAGE IN DISCUSSION**

The authors have made responses to the review concerns, please engage in the discussion,

Yours,
AC

---

### Comment · Area_Chair_bnyc · 2025-11-27

Dear reviewers,

Please review the rebuttal and discuss with the authors.

Thanks,
AC

---

### Meta-Review · Area_Chair_gBdf · 2026-01-15

**Summary:**

[AC: Overall, I do not see any ill intent during the rebuttal period. The initial ratings are 4, 4, 6, 4. The reviews are mixed. After checking carefully the author's responses and the interactions between authors and reviewers, I notice that some comments are not addressed properly. One reviewer with score 4 also indicates that his core concern is not addressed and has to decrease his score. However, one reviewer with score 4 will increase his score to 6. Given these changes, I believe this paper needs more work and is not ready to be considered for publication.]

Reviewer dmgx (4: marginally below the acceptance threshold; 3: You are fairly confident in your assessment.)

o	This paper greatly violates ICLR's formatting guidelines (see suggestions below). The paper should not be accepted in its current form.

o	Equation (10) is not rigorous. The kronecker product of  and  initialized with Equation (9) is not an identity matrix (the diagonal entries are not all 1 and there are non-zero off-diagonal entries)

[AC: The formatting issue is relatively minor. But, the authors admitted that Eq. (10) is not rigorous and have updated it accordingly. But, the reviewer did not respond to their response.]

Reviewer V2MN (4: marginally below the acceptance threshold; 3: You are fairly confident in your assessment.)
o	Can the compressed CLIP model be widely applied to downstream tasks? The experiments in this paper only evaluated retrieval and classification tasks, without assessing other tasks such as generation or some comprehension tasks. If the model can only improve performance in classification and retrieval but not be applicable to these other tasks, its applicability in a broader range of fields would be significantly reduced. It would be even better if comparisons with other compressed models on other tasks could be provided.
[AC: The authors indicate these are the current limitations of the proposed method.]

o	Diagonal initialization is one of the main contributions of this paper, which is also experimentally validated. However, the improvement compared to other initialization methods is quite significant. The diagonal initialization method has been mentioned in other works (e.g., "On the Parameterization and Initialization of Diagonal State Space Models"), and it is not a novel algorithm. Therefore, its innovation seems limited.

[AC: The reviewer indicates that he has increased the score to 6.]

[AC: Overall, these questions are not so critical.]

Reviewer TpdV (6: marginally above the acceptance threshold;  3: You are fairly confident in your assessment.)

o	This paper's core idea is that mapping preserves more information than selection currently remains at an intuitive level, lacking rigorous theoretical support. For instance, the authors fail to analyse from a mathematical perspective why this mapping-based structure can more effectively maintain the knowledge fidelity of pre-trained models.

[AC: The authors indicate that their conclusion is validated empirically. There is no theoretical proof indeed.]

o	The experiments lack ablation studies. For instance, the paper provides no ablation experiments to distinguish the respective contributions of width and depth compression.

[AC: The authors indicate that the ablation studies were provided. But, I do not see them. There is no clear indication of whether additional results were generated.]

Reviewer 24zq (4: marginally below the acceptance threshold.;  4: You are confident in your assessment, but not absolutely certain.)

o	For depth compression, it assumes the new layer can be expressed as a linear function of the old layer but it may not be true. Its primary goal is to preserve a portion of the original weights, which is already demonstrated by diagonal initialization. Can we also construct other Kronecker product matrices with a block product of 1 to preserve some weights?

o	The performance drops dramatically especially on Stanford Cars(from 88.5 to 50.8) and MNIST(from 65.8 to 13.0).

o	The results of other methods like quantization and distillation shall be compared.

[AC: The reviewer indicates that the core concern is not addressed properly, and may decrease his score. ]

**Reviewer Concerns:**

See the summary section.

**Reviewer Scores:**

[AC: Overall, I do not see any ill intent during the rebuttal period. The initial ratings are 4, 4, 6, 4. The reviews are mixed. After checking carefully the author's responses and the interactions between authors and reviewers, I notice that some comments are not addressed properly. One reviewer with score 4 indicates that his core concern is not addressed and has to decrease his score. However, one reviewer with score 4 will increase his score to 6. Given these changes, I believe this paper needs more work and is not ready to be considered for publication.]

---

### Decision · Program_Chairs · 2026-01-26

Reject